# Gene Expression Maps in Plants: Current State and Prospects

**DOI:** 10.3390/plants8090309

**Published:** 2019-08-28

**Authors:** Anna V. Klepikova, Aleksey A. Penin

**Affiliations:** 1Institute for Information Transmission Problems of the Russian Academy of Sciences, Moscow 127051, Russia; 2Skolkovo Institute of Science and Technology, Moscow 143026, Russia

**Keywords:** transcriptome map, gene expression atlas, plant, RNA-seq, microarray, database

## Abstract

For many years, progress in the identification of gene functions has been based on classical genetic approaches. However, considerable recent omics developments have brought to the fore indirect but high-resolution methods of gene function identification such as transcriptomics, proteomics, and metabolomics. A transcriptome map is a powerful source of functional information and the result of the genome-wide expression analysis of a broad sampling of tissues and/or organs from different developmental stages and/or environmental conditions. In plant science, the application of transcriptome maps extends from the inference of gene regulatory networks to evolutionary studies. However, only some of these data have been integrated into databases, thus enabling analyses to be conducted without raw data; without this integration, extensive data preprocessing is required, which limits data usability. In this review, we summarize the state of plant transcriptome maps, analyze the problems associated with the combined analysis of large-scale data from various studies, and outline possible solutions to these problems.

## 1. Introduction

Among the central objectives of biological studies are the determination of gene functions and interactions among genes in the development and life cycles of organisms. The majority of experiments aiming to identify gene function or structure and the dynamics of gene regulatory networks have been conducted on a few model objects, such as *Arabidopsis thaliana* or *Drosophila melanogaster*. Then, the obtained data are applied to other organisms. However, even for model species, only 20%–30% of genes have been the subjects of genetic experiments [1]; thereby, the utilization of additional data is essential for, at the very least, the indirect characterization of gene function. Detailed profiles of gene expression can be used for this purpose [2].

The most organized and thorough source of expression profiles is called a transcriptome map or gene expression atlas. In the paper of Schmid et al. [3], a transcriptome map was defined as a global estimation of gene expression in all possible cells, tissues, organs, or parts of an organism during the life cycle of the organism from embryogenesis to senescence. By definition, a complete transcriptome atlas requires a great deal of effort for the collection and processing of samples, which is financially costly. Nevertheless, the development of whole-transcriptome technologies in the past two decades has allowed the construction of transcriptome maps with various resolutions.

Detailed transcriptome maps can also be applied to study the genes of non-model species. Usually, gene functions in non-model species are extrapolated from the studied genes of model organisms using nucleotide sequence similarity. This approach is necessary because conducting experimental studies for all species is impossible due to costliness. Functional extrapolation of biological function based only on nucleotide similarity can be complicated. For instance, the functional transition between genes is well known for families of genes with 60% sequence identity. One amino acid substitution in the product of the flowering repressor gene *TERMINAL FLOWER 1* (*TFL1*) reverses the gene function, i.e., the results in the activation of floral transition, turning *TFL1* into the gene *FLOWERING LOCUS T* (*FT*), and vice versa [4]. The comparison of orthologous genes in model and non-model species can improve the identification of gene function in non-model objects. In the example above, the expression profiles of *TFL1* and *FT* are conservative and can be indirect proof of function [4].

Current transcriptome maps substantively differ in resolution, i.e., in the number of included organism parts, organs, tissues, or cells. A detailed transcriptome series of a single organ or process in some cases is also called the transcriptome map [5]. The current review is focused on all-organ gene expression atlases; thus, a discussion of single-organ transcriptome maps is beyond the scope of this review.

## 2. The Dawn of Plant Transcriptome Maps

The era of plant-wide transcriptomics started in 2005 when the first detailed transcriptome map was constructed for *A. thaliana* [3]. Arabidopsis is a classical object in plant biology and the first plant with a sequenced genome [6,7]; the next stage of large-scale studies on *A. thaliana* was the construction of the Affymetrix ATH1 microarray. The transcriptome map of Schmid et al. [3] became a milestone in plant transcriptomics and shaped the main areas of investigation for subsequent atlases.

Since 2000, the number of sequenced plant genomes has continuously increased across a wide range of taxa. The growing body of genomic data includes both model objects [8,9] and agricultural plants [10,11,12,13,14], as well as many other species. Progress in sequencing and assembly technologies allowed the creation of detailed transcriptome maps using both microarray and RNA-seq data.

To characterize the current state of plant transcriptomics, we have centered our review on 42 gene expression atlases (Appendix A). We have described the representation of the tree of life in the transcriptome map studies, discussed how to choose the methods of expression analysis and the most suitable samples of plant organs and tissues, and provided an overview of the common questions addressed in gene expression atlases.

## 3. Where to Look for Uninvestigated Plants?

At least 43 transcriptome maps thoroughly describe 32 plants, including many model objects or agricultural plants; therefore, reanalysis of the existing atlases can be of use in a variety of studies. In addition, 40 transcriptome maps are a drop in the ocean of plant biodiversity, encompassing more than 350,000 species only in Angiosperms and more than 30,000 species of non-seed plants [15]. We used the last variant of plant classification, i.e., Angiosperm Phylogeny Group (APG) IV, for the evaluation of species diversity and identification of the gaps in plant taxa representation [16] (Figure 1).

Only three gene expression atlases of two species represent non-seed plants: *Physcomitrella patens* of the division Bryophyta [17,18] and the gymnosperm *Pinus pinaster* (maritime pine [19]). The basal angiosperms are represented by a single species, avocado (*Persea americana*), which belongs to the family Lauraceae [20]. More than one-third of the 43 transcriptome atlases describe monocots from only one family, Poaceae, which is expected because grasses are among the main agricultural plants. In 2018, the Food and Agricultural Organization (FAO) estimated the yield of cereals was 2652 million tons [21]. The studied plants include wheat (*Triticum aestivum*) [22,23], rice (*Oryza sativa*) [24,25], maize (Zea mays) [26,27], sorghum (*Sorghum bicolor*) [28], and barley (*Hordeum vulgare*) [29].

Dicots are represented by 24 atlases, and, similar to monocots, the distribution of dicot taxa is biased. Only four Asterids species are represented by the transcriptome maps: Coffee (*Coffea arabica*) from Rubiaceae [30], tobacco (*Nicotiana tabacum*) [31], potato (*Solanum tuberosum*) [32], and tomato (*Solanum lycopersicum*) [33] from Solanaceae. One of the main web expression visualizers, eFP browser [34], includes the transcription profiles of tomato and potato (Solanaceae); however, in this study, the expression data were used mainly for genome annotation and were not analyzed in detail, so we did not consider these profiles in our review [35,36]. Although relatively intensive studies have investigated rosids, the diversity of the investigated families is still poor: To our knowledge, transcriptome maps are available for species from Brassicaceae, Cucurbitaceae, Fabaceae, Rosaceae, and Salicaceae (Figure 1). The Fabaceae family (11 transcriptome maps) includes many agricultural plants, such as peanut (*Arachis hypogaea*) [37], alfalfa (*Medicago sativa*) [38], soybean (*Glycine max*) [39,40], and chickpea (*Cicer arietinum* [41]). The classical model object *A. thaliana* [1,3], the model of the early evolution of polyploids *Capsella bursa-pastoris* [9], and agricultural species *Brassica rapa* and false flax (*Camelina sativa*) [42,43] belong to Brassicaceae (all found in five atlases).

Thus, several families have been well studied, and many model and agricultural plants are already covered. More than one transcriptome map has been constructed for several species (for instance, Arabidopsis, sorghum, or rice), reflecting the common practice of updating the previous datasets, while for novel transcriptomic data, one should look outside the popular families. Gene expression atlases of uninvestigated taxa are especially important for evolutionary studies, where a biased set of analyzed species can lead to erroneous conclusions [44,45]. The investigation of basal angiosperms such as *Amborella trichopoda* or *Nymphaea caerulea*, the non-grass monocots (e.g., bananas, date palm, or the orchids), basal eudicots, and many families from superrosids and superasterids will correct this imbalance and improve the understanding of the diversity of the plant transcriptome and evolution of expression patterns in plants.

## 4. What Are the Modern Strategies of Transcriptome Map Construction?

With the exception of a study by Nobuta et al. [24] in which 18 samples of rice were analyzed using massively parallel signature sequencing (MPSS), all of the reviewed atlases were created with microarrays or RNA-seq—the most typical methods used in transcriptomics.

Fourteen out of the 43 gene expression atlases were constructed using microarrays. With all known limitations of microarrays (preset number of genes and isoforms, low dynamic range, difficulties with paralogous genes [46]), clear protocols and quality standards of microarrays [47] make them attractive methods even in the present day. The resolution of a microarray (the number of genes on a given microarray) is dependent on the genome assembly and annotation quality, so the progress of genome technologies increases the completeness of microarrays. Among the studied transcriptome maps, the percentage of genes on a given microarray varies from 72% (maize [26]) to 100% (wheat and sorghum microarrays [22,48]).

Although microarray-based transcriptome maps are still being published (Appendix A), the number of maps published every year has decreased with time, and they are being substituted with RNA-seq-based atlases (Figure 2). The direct comparison of maize transcriptome maps based on microarray and RNA-seq showed good repeatability of the results, though transcript detection was substantially lower for microarray, limiting its application [26,46]. Twenty-seven of the observed transcriptome maps were constructed using RNA-seq. To our knowledge, the first RNA-seq-based atlases were collected for rice (8 samples [49]) and soybean (9 and 14 samples [39,40]).

During the history of next-generation sequencing, many technologies rose and fell, bringing their own advantages, biases, and limitations. Currently, the Illumina (San Diego, CA, USA) technology of sequencing by synthesis with reversible terminators dominates the area of transcriptome maps: Only three atlases were constructed using SOLiD (version 3.0 and 4.0, Applied Biosystems) [20,50] and GS-FLX+ (Roche Applied Sciences (Indianapolis, IN, USA)) [19].

The balance of desirable atlas size and experimental cost in RNA-seq-based transcriptome maps is achieved primarily by the selection of read length and sequence depth. Read length influences the mapping quality, which is not crucial for species with well-assembled and annotated simple genomes but is essential for polyploids and other variants of complex genomes. In recent studies, read length is typically 100 bp [51,52] and may be up to 150 bp [28] (Appendix A). Paired-end reads facilitate successful mapping and have been used in the majority of atlas studies. However, for plants with high-quality genome assemblies, such as *A. thaliana* and maize, single reads have been effectively used [1,52], which allows the analysis of more samples and/or increases in sequencing depth because of cost reduction.

Increases in sequencing depth enable a more precise analysis and lead to robust conclusions regarding gene expression. In the studied maps, the minimum sequencing depth, 9.3 M reads, was observed in the rose atlas [53], and the maximum sequencing depth, 4.5 B reads, was observed in maize [27] (Appendix A). For individual samples, the suitable library size depends on the aim of the study: Several million reads are sufficient for the transcriptome of a single cell, while a splicing analysis for a selected organ requires approximately 100 M reads [54].

Increases in sequencing depth enable a more precise analysis and lead to robust conclusions regarding gene expression. In the studied maps, the minimum sequencing depth, 9.3 M reads, was observed in the rose atlas [53], and the maximum sequencing depth, 4.5 B reads, was observed in maize [27] (Appendix A). For individual samples, the suitable library size depends on the aim of the study: One million reads are sufficient for the transcriptome of a single cell [54] with as few as 50,000 reads enough to reliably detect the transcripts of highly expressed genes [55], while a splicing analysis for a selected organ requires approximately 100 M reads [54].

Currently, the majority of transcriptome maps were created using the extraction of polyadenylated RNA with oligo-dT-beads, which leads to the reliable detection of the expression of most protein-coding genes and some noncoding RNAs. The composition of small RNAs (smRNAs) was analyzed in the MPSS-based atlas of rice cells [24], and rRNA depletion was used in the gene expression map of clover (*Trifolium subterraneum*), though noncoding RNAs were not analyzed in detail in the study [56].

Thus, present transcriptome maps are limited by the Illumina platform, which has great performance in terms of gene expression level estimation but is not suitable for the detection of alternative transcript variants. The application of polyA extraction protocols for sequence library construction leaves non-polyadenylated noncoding RNAs, both small and long, beyond consideration; strand-specific protocols, which allow determination of the expression of antisense RNAs, are rarely used in plant transcriptomics either.

The currently used methods of gene expression atlas construction provide a considerable scope for the application of various sequencing technologies, such as linked-read sequencing, synthetic long reads, Hi-C, and Nanopore (Oxford Nanopore Technology, Great Britain).

## 5. How to Choose the Representative and Comparable Set of Samples?

Gene expression atlases vary widely by resolution, i.e., the number of samples, which is influenced by the aim of study, year of creation, and technology used. Among the maps included in this review, the minimum number of samples was five, and the maximum number of samples was 79 (Figure 1). The basic set of examined samples reflects the general plant morphology and is similar among all atlases. This set usually includes ripening and mature seeds and/or fruits and/or embryos, germinating seeds, seedlings (or their parts), stems, leaves (typically more than at one developmental stage and in parts), flowers (usually separated organs), and roots. Using the set as the basis of a planned transcriptome map significantly increases comparability with previous atlases, as was shown for wheat and tomato [22,33].

The extension of a basic sample set is dependent on the aim of the study. For instance, for the cultivated monocots (barley, rice, and maize), samples important for productivity—inflorescence, floral organs, and endosperm—have been explored in-depth [23,25,26,27,29,46,49]. Legume crops are used for the nitrogen enrichment of soil, which is essential for sustainable agriculture, so maps of soybean and barrelclover (*Medicago truncatula*) were centered on roots and nodules (the structures containing symbiotic nitrogen-fixing bacteria) [39,40,57]. For the oilseed crop Camelina, seeds and flowers were analyzed in detail in the transcriptome map [43].

It is important to consider that transcriptome map consisting of whole organ samples has lower resolution than gene expression atlas comprising tissues or cell types: The number of genes expressed in at least one of the floral organs (sepal, petal, stamen, or carpel) was 11% higher than that expressed in the whole flower [3]. Similar observations were made in the rice cell atlas, where in many cases, the gene expression observed in certain cell types was not detected in whole organs [58]. Another common practice is to complement a developmental transcriptome map with samples subjected to stress treatments, both biotic and abiotic [1,24,32,53].

Existing transcriptome maps can be used for the sample selection of a new atlas, as shown in barley, Capsella, and tomato maps [9,22,33]. The careful choice of the most divergent samples is based on the analysis of transcriptional relatedness of different organs and tissues of plants. The common method of such analysis is the clustering of samples based on gene expression profiles.

Despite the differences explained by individual plant characteristics, the clustering of samples is similar in various transcriptome maps and represents the morphological and physiological congruence of organs. In general, samples are clustered by function, i.e., similar organs at different developmental stages are grouped together, and different organs collected from plants at certain ages are not grouped together. Samples are clustered by the presence of chlorophyll (which was the main factor for barley [29]) and by the content of certain tissues (samples with meristematic tissues are organized in clusters in the atlases of *A. thaliana* and maize [1,27]). Though clustering by age is not common, senescent organs form separate clusters, which can be explained by the dominance of desiccation and tissue decay processes over tissue-specific determinants [1,3].

Using high-resolution gene expression atlases, the precise clustering of anthers, different parts of leaves, and green internodes, all types of roots, seeds, flowers, floral organs, and meristems can be shown [1,25,27,28,48]. Such clades also appear in smaller maps, in which roots, flowers, seeds, and the green parts of plants usually form groups [39,40,57,59].

The analysis of clustering revealed significant differences in the pollen of different samples in *A. thaliana*, rice, and maize [1,24,25,27]. Although seeds clustered together, there were significant expression differences between seeds in detailed seed development series for *A. thaliana* and maize, which emphasizes the need for precise seed developmental stage choices in studies of related processes, especially for crops [1,26].

## 6. Overall Characteristics of the Plant Transcriptome

Though the plants described by gene expression atlases belong to a variety of taxa, the number of expressed genes is quite uniform across the majority of studies (Appendix A). The percentage of genes expressed in at least one sample varied from 37% to 93%, and most maps had a percentage in the range of 75%–85%, with no dependence on taxa (Appendix A); the extremes of the distribution are explainable by the methods of study.

In two studies, a control to determine expressed gene detection completeness was performed. In soybean samples, 74.5% of genes were expressed in at least one sample, and after the addition of 14 root samples from another study [60], the percentage of expressed genes increased to 76.5% [39]. The inclusion of stress-treated samples led to the expression of only an additional 96 (26 protein-coding) genes in the *A. thaliana* map [1]. Thus, the correct choice and a sufficient number of samples allow the detection of almost all potentially transcribed genes.

In most cases, samples show little difference in the number of expressed genes. In general, the difference between samples with the minimum and maximum numbers of expressed genes is 10%–20%, and the percentage of detected genes is 50%–70% (Appendix A). 

The trend in which the majority of genes are expressed in all or almost all samples has been observed for various species, with 50%–73% of gene transcripts present in almost all samples [1,3,43]. Additionally, in detailed transcriptome atlases, a second gene number peak was observed in a few samples. Thus, the overall expression patterns of genes show that the majority of genes are universally expressed and that there are also genes specific to certain organs and stages. Genes expressed in an intermediate number of samples are specific to groups and series of samples in transcriptome maps.

The number of genes expressed in all samples varied with a range of 18%–50% between the transcriptome atlases, although the majority of percentages ranged between 42% and 50% (Appendix A). This observation is unlikely to be the consequence of study sample number because the resolution of the gene expression atlases varied between 6 [42] and 60 samples [26].

Analyses of non-expressed genes are complicated because insufficient sequencing depth or the absence of specific genes in a microarray can make transcript detection impossible. We found two typical values of non-expressed gene ratios: Approximately 10% (8.6% for the maize atlas and 12% for the *A. thaliana* and *C. sativa* atlases) [3,26,43] and 22%–23% [1,39,57]. The bimodal distribution of non-expressed gene number point to the lack of understanding of what percentage of genome is silent among the plants.

These parameters are different at the level of cell type. The cell atlas of rice represents a microarray analysis of the expression profiles of 30,731 genes in 40 cell types [58]. A total of 80.8% of the genes were expressed in at least one sample, and the expressed gene ratios of different cell types showed pronounced differences, ranging from 26% to 52%. Only 7% of genes were expressed in all samples. In contrast with other transcriptome maps that include whole plant organs or tissues, the transcripts of the majority of genes were detected in a few cell types, and various cell types had minor expressed gene intersections. This result is evidence that expression variability differs at various levels of organization (cells, tissues, or organs), which should be considered in the design of transcriptome maps.

Another overall characteristic that is often evaluated in transcriptome maps is expression profiles of samples. Despite a similar number of expressed genes in the samples, the overall expression profiles in organs and tissues vary greatly [3]. The Z score was initially suggested as a suitable measure for the general transcription profiles of samples in the *A. thaliana* transcriptome map [3] and has since been used for a variety of objectives. The Z score describes the deviation of the gene expression in a given sample from the mean expression across all samples of the atlas. The transcriptome map structure (number and selection of samples) directly influences the mean gene expression; thus, global transcriptional profile characteristics of samples can significantly differ among species.

The tissue with the closest to mean expression profile in some species can be leaf (*A. thaliana*, *M. truncatula*, and maize [3,26,57]) or seed (soybean [40]), but in other plants, the distribution of Z score in these samples can be wide or even bimodal (*Panicum hallii*, *M. truncatula* [50,56]) as well as in flowers of *A. thaliana* [3] or soybean roots and nodules [40].

## 7. What Questions Can Be Addressed by a Gene Expression Atlas?

Though the 43 transcriptome maps have their own research aims, several questions are addressed in many of them, so we highlighted these topics—stable genes, tissue specificity, transcription factors, functional analysis, alternative splicing, evolutionary studies—in the review below.

### 7.1. Stably Expressed Genes

The identification of genes with an expression level that remains constant across tissues, organs, and conditions (stable genes) is very important for the normalization of gene expression among samples. Stable genes are widely used as internal references in expression comparisons using real-time qPCR [61]. The identification of uniformly expressed genes has been conducted for many plants [62,63,64,65,66] (for review, see [67]), and transcriptome maps are suitable for providing information about gene expression in many tissues, organs, and conditions.

Coefficient of variation (CV), which is the standard deviation divided by the mean across all samples, was used as a measure of expression stability in the majority of the papers. Notably, increases in the number and variety of analyzed samples lead to decreases in the number of stable genes, and the possibility of finding a sample in which stable genes have significantly different expression levels cannot be eliminated.

Depending on the CV threshold, which in most cases is in the range of 0.1–0.3, the number of stably expressed genes can vary from a few dozen [1,38,50] to almost 1000. The majority of these genes have not been annotated with any known function [32,57]. The Gene Ontology (GO) enrichment results of stable genes differ among plants, although in all cases, categories associated with processes essential for life are overrepresented.

The traditional references GAPDH and ubiquitin have been identified as stable in several transcriptome maps [25,50,57,68]; however, classical housekeeping genes are usually not in the list of most stable genes and have high dispersion values up to CV = 0.96 [1,25,26]. Stable genes from Czechowski [61] were verified in the transcriptome atlas of *A. thaliana* and have CV < 0.3; thus, these genes can be used for precise normalization [1] (Figure 3).

### 7.2. Tissue-Specific Expression Analysis

The identification and analysis of genes expressed in certain tissues and organs help to answer questions about the mechanisms underlying the development of unique morphological and physiological features. Thus, searching for tissue- and organ-specific genes is a key issue in transcriptome map analysis. The identification of unique organ and tissue genes can improve the understanding of the mechanisms underlying the uniqueness of each organ and has a practical application: The precise choice of target genes for genetic manipulations can increase economic performance [26]. The promoters of tissue-specific genes can be used for the design of transgenic constructs specifically expressed in certain organs at definite stages of development, which can reduce the negative effects of constitutive transgenic expression [69].

Two main approaches can be used for tissue-specific gene identification. The analysis of differential expression (DE) between all possible pairs of samples and the identification of genes expressed mainly in one sample allow the discovery of tissue-specific genes together with the evaluation of general transcriptome diversity. This method has been applied to a variety of objects [17,38,50,51,59,70]. The number of DE genes varies from 14 (stages of meristem development in *A. thaliana* [1] to 27,945 (between the seeds and flowers of soybean [40]), which reflects significant differences in the transcriptional profiles of organs and tissues.

After the identification of DE genes, the expression level fold change can be used for gene clustering and the identification of tissue-specific genes, as performed for *P. hallii* [50], black cottonwood (*Populus trichocarpa*) [70], pea (*Pisum sativum*) [59], and Camelina [43].

The second approach to tissue specificity is the employment of numeric metrics of expression pattern width. The binary approach is the simplest version: For each sample, a given gene is determined to be expressed or not expressed, and expressed genes identified in only one sample are called tissue specific. Approximately 1000 tissue-specific genes have been identified using a binary approach for several objects in *M. truncatula* [57], common bean (*Phaseolus vulgaris*) [51], tobacco [31], cowpea (*Vigna unguiculate*) [69], and maize [26].

The Z score used for the general assessment of sample transcription profiles can also be applied for the identification of tissue-specific genes. The maximal Z score corresponds to the maximal deviation from the mean of the expression of a given gene in a given sample. Genes unique to soybean nodules and seeds were discovered using the Z score [40], and in another legume, *Lotus japonicus*, nodules also contained the highest number of unique genes discovered with the Z score (29% of all 2949 unique genes) [68].

The specificity measure (SPM) was used to search for unique organ and tissue genes in *Brachypodium distachyon*. This measure varies from 0 if a gene is not expressed in the sample to 1 if the gene is expressed exclusively in the sample [71,72]. SPM was successfully used for the identification of several thousand tissue-specific genes in the seeds, leaves, and internodes of *B. distachyon* [71].

Shannon entropy H is a commonly used measure of tissue specificity [73,74]; a high value of H corresponds to ubiquitously expressed genes, while genes unique to tissues have low values of H. Shannon entropy distributions are strongly skewed to the right, indicating that the majority of genes are expressed in all or almost all samples, and the second pronounced spike in gene number that occurs at low values of H consists of tissue-specific genes [1,46]. Using Shannon H, rice genes specific to the flower, endosperm, roots, and callus were identified [25].

### 7.3. Detailed Studies of Transcription Factor Expression

Among tissue-specific genes, transcription factors (TFs), which are DNA-binding proteins regulating the activity of other genes, are particularly valuable for establishing the unique properties of tissues and organs. The orchestrated regulation of gene expression is essential for correct biological processes, and TFs are the key nodes in gene regulatory networks [43]. TF families are highly conserved among eukaryotes [75], which helps with comparisons of transcriptome maps of plants from different taxa.

The tissue specificity of several TF families, including C2H2 (Zn), LIM, MADS, Myb-R2R3, ERF, and NAC, was studied in soybean, tobacco, and maritime pine, allowing the inference of previously unknown functions of TFs [19,31,39].

TFs regulating sporophyte development via the negative regulation of gametophyte development were the focus of an analysis of *P. patens* moss. The transcription of these TFs was specific to the apical meristem, and genes encoding proteins that are known to interact were expressed at similar levels [17].

The tissue specificity of TFs is studied mainly at the expression level, so the combination of transcriptome maps with other methods, such as ChIP-seq, can be a valuable source for gene expression network inference.

### 7.4. Functional Analysis of Genes Using Transcriptome Maps

Understanding the molecular and physiological processes underlying studied phenomena is the main aim of biological research. The majority of the papers in this review conducted functional analyses of biological processes and the quantitative characterization of transcriptome maps. Many plants that were the subjects of these transcriptome maps are model systems or economically important, and the organs or tissues involved in related processes are studied with a high morphological or temporal resolution.

In the observed transcriptome maps, attention was mainly paid to symbiosis with nitrogen-fixing bacteria (*M. truncatula*, soybean, pea [39,57,59]), lignin biosynthesis (maize, alfa alfa, Panicum [26,38,50]), and fruit development (Camelina, *V. unguiculata*, avocado, and grape [20,43,69,76]), which correlate with the agricultural value of plants. The analysis of transcriptome maps revealed the key regulators of these processes and control of associated metabolism. Additionally, genes associated with leaf development were identified for tobacco [31]; in the transcriptome atlas of rose orthologues of known genes, regulated transition to flowering and the development of meristem and floral organs were analyzed [53]; and nitrogen metabolism was studied in maritime pine [19].

### 7.5. Analysis of Evolutionary Processes

Orthologous genes presumably retain gene function; this should lead to similar expression patterns in different plants [43]. The accumulation of expression data, including transcriptome maps, allows analysis of the conservation of expression patterns among plants with different evolutionary distances.

The expression patterns of orthologous genes of Camelina and *A. thaliana* from Cruciferae are typically similar [43]. A comparison of expression patterns over a greater evolutionary distance was conducted for homologues of *M. truncatula* and *A. thaliana* using the transcriptional profiles of comparable organs from the respective transcriptome maps [3,57]. The similarity of expression was measured using the Pearson correlation coefficient, and expression patterns were retained for 62% of the orthologues, indicating that a significant number of orthologues showed evolutionary functional changes [57].

The availability of several legume transcriptome maps provides the opportunity to study genes unique to Fabaceae. Genomes of soybean, barrelclover, and *L. japonicus*, which diverged 40 Mya, retain microsynteny. The expression of 40 tissue-specific soybean orthologues has been analyzed in the transcriptome maps of *M. truncatula* and *L. japonicus* [57,77]. The expression patterns of 18 triplets of orthologous genes were retained, while the other 22 triplets were either non-orthologous or in the process of neo/subfunctionalization. These 22 genes are in syntenic regions; thus, divergence represents the more plausible explanation [39]. A high level of tissue specificity of legume-specific genes was also shown in the transcriptome map of *L. japonicus* [68].

In a paper by Vlasova et al. [51], in addition to the transcriptome map of the common bean phylome, the complete evolutionary history of *P. vulgaris* genes was constructed. The phylome was used for the analysis of the divergence of expression patterns of genes formed in earlier duplication events [51].

Transcriptome atlases of polyploids have been used to study the expression differences between homoeologues. For allotetraploid *C. bursa-pastoris*, the analysis of differential expression between homeologues showed the absence of asymmetrical silencing of one subgenome [9]). The analysis of DE among homoeologous triplets of the allohexaploid Camelina revealed that the overall expression of two genomes was equal, while the third genome had higher expression [43].

### 7.6. Analysis of Alternative Splicing

Alternative splicing (AS) is one of the mechanisms of gene expression regulation that influences the complexity, quantity, and stability of transcripts [78]. As we discussed above, most RNA-seq-based transcriptome maps are constructed using the Illumina platform and relatively short reads, which limits the possibility of splicing analysis. However, several studies have shown that alternative isoforms are functional in different plant organs.

In the analysis of AS events in *A. thaliana*, strict criteria for splice junction detection were used to avoid mapping artefacts [1]. A total of 221,187 splice junctions were identified, and only 115,686 of these junctions were present in the TAIR10 *A. thaliana* genome annotation. Notably, splice junctions annotated in TAIR10 were usually found in almost all samples, while newly discovered junctions were found in only a few samples [1].

A total of 276,275 AS events in 39,620 genes were identified in Camelina [43]. The most common event was intron retention, which is concordant with previous discoveries in other plants. The distribution of different AS events was similar in each sample when the frequency of AS was higher in younger tissues. Genes encoded splicing factors were DE among many samples, which can be attributed to tissue-specific AS.

A similar number of AS events (204,503) was found in the transcriptome map of peanut [37], and, as in the case of Camelina, the ratio of events did not differ among tissues.

## 8. How to Summarize the Data–Expression Databases

To improve the usability of transcriptome maps, specific databases are constructed in many studies. Such databases can be divided in two categories with different toolkits: First, those aimed at user-friendly visualization, and second, those that allow additional analyses.

If the main aim of the database is convenience, data displaying only images of plants with depicted gene expression levels are available for users (Figure 4a). One of the most well-known resources of this type is the eFP browser (for Arabidopsis, see http://bar.utoronto.ca/efp/cgi-bin/efpWeb.cgi) and its extension ePlant (http://bar.utoronto.ca/eplant), which includes other data types (such as chromosome position or phylogeny) in addition to gene expression [34,79]. Among other such databases as AgriSeqDB (https://expression.latrobe.edu.au/agriseqdb [80]) or TodoFirGene (http://plantomics.mind.meiji.ac.jp/todomatsu/ [81]) can be named.

The second type of database provides a variety of tools for user analysis with limited visualization because the simultaneous presentation of a high amount of data is complicated. One example of such a resource is the transcriptome variation analysis (TraVA, http://travadb.org/ [1]) (Figure 4b) database. In this case, the possibility of partial data reanalysis is a central focus. Users can perform a chosen type of expression normalization such as RPKM, FPKM (Reads or Fragments Per Kilobase Million), and other variants (PLEXdb, http://www.plexdb.org, containing the transcriptome maps of barley, maize, and rice [25,26,29], EBI Expression Atlas (https://www.ebi.ac.uk/gxa/plant/experiments), or TraVa), the simultaneous presentation of several genes (Melonet-DB, http://gene.melonet-db.jp/cgi-bin/top.cgi [82]), DE analysis (https://mtgea.noble.org [57,83] and TraVA), coexpression analysis (https://mtgea.noble.org, PLEXdb, TomExpress http://tomexpress.toulouse.inra.fr [84]), or other functions.

## 9. Where to Go Next—Future Perspectives of Transcriptome Maps

Our review of transcriptome maps shows the exceptionally high level of technology development in this field. However, a significant number of problems remain unsolved in plant transcriptomics. One key issue is the pronounced taxa representation bias of gene expression atlases. The lack of coordination in transcriptome map construction complicates subsequent analyses.

The second direction essential for extensive data usage is the development of analytical database interfaces that help users without a bioinformatics background to handle the results of multiple experiments, i.e., to compare expression profiles, identify coexpressed genes, change the parameters of analysis, and so on. Unfortunately, at this moment, most databases either apply analyses and provide technical information in an inconvenient format or create attractive figures that do not support the ability to influence results. These databases also do not allow the addition of new information by users. The development of a database that is open for uploads and has an interface integrated with a large amount of data is a highly complicated task, but this task is important for the proper usage of large bodies of data.

The exclusive focus of experiments on protein-coding genes also represents a problem. Expression analyses of smRNAs and long noncoding RNAs have been conducted in a few studies [24,37], but these studies do not encompass the full species diversity of plants, while for animals, atlases of noncoding RNAs already exist [85]. Although AS is addressed in several studies, the majority of these studies have insufficient data for robust conclusions.

The further progress of novel methods such as single-cell approaches and long-read sequencing will help to produce plant transcriptome maps with better resolution. These methods have already proven to be useful for animals, e.g., a single-cell map of the Drosophila brain [86] and a long-read gene expression atlas of rabbit are available [87]. Despite several problems with the adaptation of methods for plants, third-generation technologies are highly promising for deeper AS inference and the improved resolution of plant cell types and dynamics, which was successfully shown on single-cell atlas of Arabidopsis root [88]. The new dimension of transcriptomics that was added was spatial expression analysis [89,90]. Additionally, for the efficient investigation of a global plant transcriptome, the worldwide coordination of studies and (meta)data sharing are highly important.

## Figures and Tables

**Figure 1 plants-08-00309-f001:**
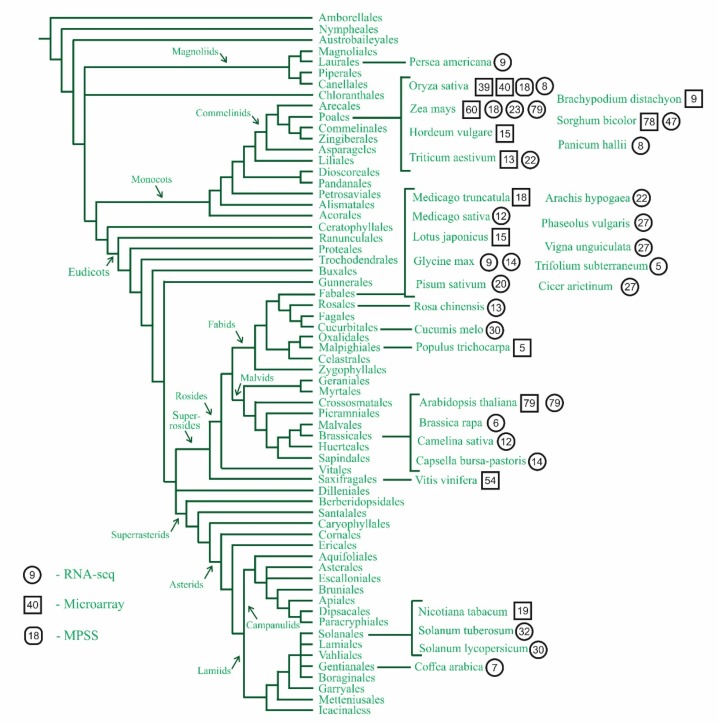
The representation of taxa among gene expression atlases. The number of samples (organs, tissues, or cells) is indicated by circles for RNA-seq-based transcriptome maps and squares for microarray-based transcriptome maps. The figure is represented the APG IV classification of plants and is adapted with permission from [16].

**Figure 2 plants-08-00309-f002:**
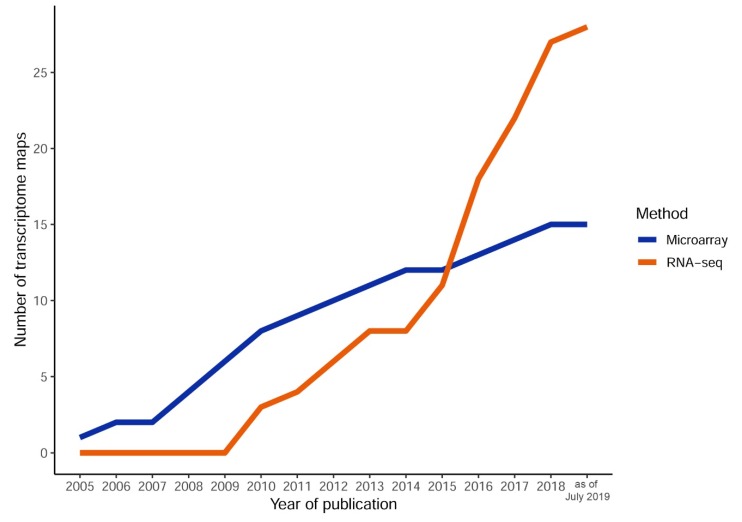
The accumulation of published transcriptome maps by year. After 2015, the number of RNA-seq-based atlases surpassed the number of microarray-based maps, which significantly decreased. The data for 2009 cover the period January to July.

**Figure 3 plants-08-00309-f003:**
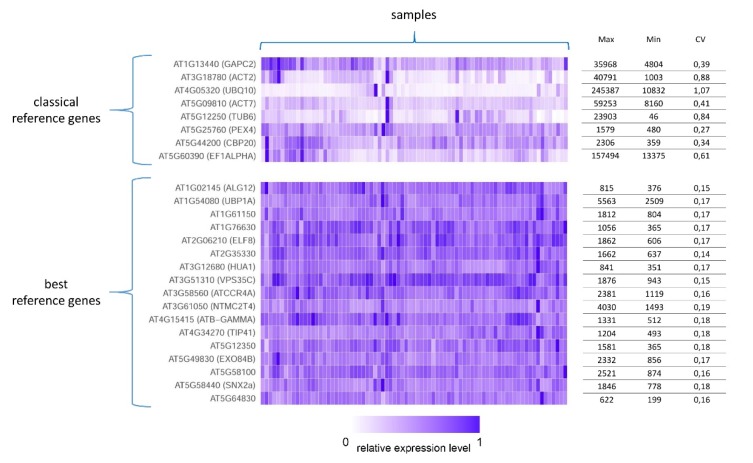
Heatmap of the expression of the most stable genes and classical reference genes in the *A. thaliana* transcriptome map. Each column of the heatmap represents a sample from an RNA-seq-based *A. thaliana* gene expression atlas [1]; all samples of the transcriptome map were used. For each row (representing genes) the maximum (Max), minimum (Min), and coefficient of variation (CV) of expression are indicated on the right.

**Figure 4 plants-08-00309-f004:**
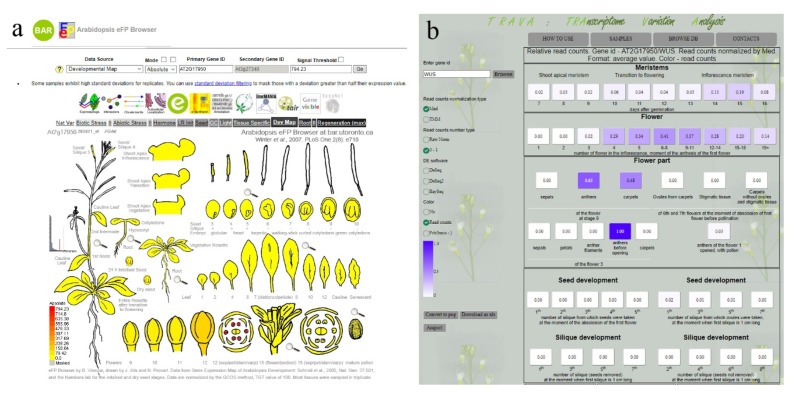
General view of expression databases. (**a**) First type (visualization-oriented) of database, eFP browser (http://bar.utoronto.ca/efp/cgi-bin/efpWeb.cgi). (**b**) Second type (analysis-oriented) of database, transcriptome variation analysis (TraVA) (http://travadb.org).

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
