# Peer review of "Gene Expression Maps in Plants: Current State and Prospects"

_plants, 2019, doi:10.3390/plants8090309_

Round 1
Reviewer 1 Report
The Review by Klepikova & Penin summarises most of the efforts undertaken so far to create transcriptomics atlases in plants by using microarray and RNA-seq. Specifically, they analyze the different atlases under different perspectives in order to facilitate their comparison/complementarity. Moreover, they underline the bias in the number and type of species for which atlases are available.
The Review is well written and will serve as a nice introduction or overview to the available resources currently available.
Major comments:
Line 152-153: be more explicit in the unit of measurement (i.e. reads, bp, etc.).
Line 155: where is the sequencing depth at single-cell level coming from? FOr example, 10x Genomics reccomends around 50,000 read-pairs per single cell (https://kb.10xgenomics.com/hc/en-us/articles/115002022743-What-is-the-recommended-sequencing-depth-for-Single-Cell-3-and-5-Gene-Expression-libraries-).
Line 470-473: more efforts in increasing the resolution of the transcriptomics atlases are ongoing. The last paragraph, "9. Where to Go Next – Future Perspectives of Transcriptome Maps" would benefit of discussing the Spatial Transcriptomics efforts (Giacomello et al., Nature Plants 2017, Giacomello and Lundeberg, Nature Protocols 2018), which aim to add a spatial resolution to the transcriptome information, as well as the single-cell efforts (Shulse et al., Cell Reports 2019).
Minor comments:
Line 75: 40 should written as number
Line 102: write Capsella bursa-pastoris - it is mentioned for the first time
Line 132: missing “)”
Lines 184-185: G. max as full name - it is mentioned for the first time Line 205: missing “)”
Reviewer 2 Report
The major concern about last para (Where to Go Next – Future Perspectives of Transcriptome Maps) of this manuscript-
There are several user-friendly databases are available which allow users to analyze omics type of datasets with customized parameters; e.g., EMBL-EBI expression atlas, PLANEX, etc. Most of the such databases offer community engagement and allow users to share their data for improvement of the existing database content. Database development team members usually take care of data upload and integration in large databases so user engagement for data upload and integration with existing data may create a mess for them. Users can send the data to them by using query form or email to contact person would be the better approach to submit data in existing databases. Users can also provide suggestion for database improvement by using feedback forms; e.g., Gramene (http://www.gramene.org/feedback). There are also several databases covering various plant species for lncRNA and AS genes, for example- PNRD (http://structuralbiology.cau.edu.cn/PNRD/index.php), RNAcentral (https://rnacentral.org), Plant Alternative Splicing Database (http://proteomics.ysu.edu/altsplice/), ERISdb (http://lemur.amu.edu.pl/share/ERISdb/home.html), etc. So, I think, second and third para (line 456-469) of last section of this manuscript need to be revised.
Other concerns -
Mention the distance between nodes in figure 1. Provide the scale for the heat map (figure 3) and brief description about the type of samples used to construct the figure and their sources with citation. The subheading “How to choose the best sample?” doesn’t go well with the content under this section so I suggest changing the subheading according to content under it. Another subheading “what is the method of choice for transcriptome map construction?” doesn’t describe the method but the strategies used for construction of gene expression map based on published literature. Again, I suggest changing the subheading according to content under it.
Reviewer 3 Report
[See the attached Word file.]
